## Research Article

integrated care; collaborative care; serious mental illness; primary health care; South Africa

**Corresponding author:**
Saira Abdulla;
Email: saira.abdulla11@gmail.com

# Healthcare providers' experiences of community-based collaborative care for serious mental illness: a qualitative study in two integrated clinics in South Africa

Saira Abdulla[1] , Lesley Robertson[2,3] , Sherianne Kramer[1] and Jane Goudge[1]

[1]Centre for Health Policy, School of Public Health, University of Witwatersrand, Johannesburg, South Africa; [2]Department of Psychiatry, University of Witwatersrand, Johannesburg, South Africa and [3]Community Psychiatry, Sedibeng District Health Services, Sedibeng, South Africa

## Abstract

Community-based collaborative care (CBCC) is an internationally recognised model of integrated care that emphasises multidisciplinary teamwork and care coordination. In South Africa, community psychiatry has been integrated into some primary healthcare (PHC) facilities. This study examines healthcare providers' perceptions of collaboration and its challenges in various integrated care settings. Three main components of CBCC (multidisciplinary teams, communication and case management) were explored through qualitative interviews with 29 staff members in 2 clinics. In Clinic-1, community psychiatry services operate independently in an outbuilding behind the main PHC clinic ("co-located"). In Clinic-2, these services are fully integrated within the PHC clinic ("physically integrated"). Both clinics had multidisciplinary teams, with various staff members conducting case management functions on an *ad hoc* basis. The physically integrated clinic (due to shared files, physical proximity and a facility manager with mental health experience) had greater levels of communication between the multidisciplinary team. In contrast, the co-located clinic struggled with poor management, unclear reporting structures and reinforced traditional hierarchies, limiting collaboration between the staff members. Integration does not guarantee collaboration. Improving collaboration between mental health and PHC staff requires clear roles, competent managers, CBCC endorsement from PHC clinicians, sufficient human resources and systematic communication channels, such as case review meetings.

## Impact statement

In South Africa, national policy mandates the integration of mental health services into primary healthcare (PHC) settings. However, while mild and moderate mental health conditions have been prioritised, people with serious mental illnesses (SMIs) remain underserved. In Gauteng province, community psychiatry services are integrated in some PHC clinics to deliver specialised psychiatric care that was once limited to institutional settings. However, without standardised approaches to integrating care, mental health services are integrated in different ways. Community-based collaborative care (CBCC) is an integrated care model that aims to foster teamwork among PHC and mental health staff. However, little is known about the impact that the type of integration has on collaboration between healthcare providers. This study, therefore, explores healthcare providers' perceptions of the extent of CBCC and the challenges it poses by comparing integration approaches. The findings show that the type of integration approach plays an important role in facilitating collaboration between staff. Furthermore, factors such as leadership, working environments and role clarity play an important role in facilitating collaboration. The findings also suggest that the integration of multidisciplinary teams does not guarantee collaboration; however, physical integration provides better opportunities for collaboration. This study provides important insights that need to be considered for policymakers and researchers when implementing integrated CBCC in resource-constrained settings.

## Introduction

Low- and middle-income countries (LMICs) have seen a significant increase in the burden of serious mental illnesses (SMIs) (Charlson et al., 2018). Despite this, the mental health treatment gap remains high (Docrat et al., 2019). To reduce the treatment gap and improve access to quality

mental health services at the community level, the World Health Organisation recommends the provision of mental health services at the primary healthcare (PHC) level through initiatives such as the Comprehensive Mental Health Action Plan 2013–2030 and the Mental Health Gap Action Programme (World Health Organization, 2016; WHO, 2021). There are no standardised approaches to integrating mental health services in PHC, and integration approaches vary based on resource availability, infrastructure and local context (Thornicroft et al., 2019).

Community-based collaborative care (CBCC), an internationally recognised model of integrated care (Whitfield et al., 2023), aims to improve healthcare user (HCU) outcomes through teamwork and care coordination among multidisciplinary teams (Ee et al., 2020). Effectiveness reviews for common mental disorders in high-income countries have shown that CBCC improves the management of mental illness and associated multimorbidities, thus improving HCU outcomes (Neumeyer-Gromen et al., 2004; Gilbody, 2006; Archer et al., 2012; Kappelin et al., 2022). However, SMIs are more challenging, and when poorly managed, SMIs often lead to psychiatric emergencies that are costly to the health system (Docrat et al., 2019). A recent global effectiveness review for SMIs found limited high-quality evidence supporting CBCC as more effective than standard care (Reilly et al., 2024). However, findings from lower-quality studies in the review (Salman et al., 2014; Mishra et al., 2017) suggest that CBCC could improve mental health symptoms and quality of life, and reduce psychiatric admissions (Reilly et al., 2024).

In Gauteng province (South Africa), community psychiatry services are provided from selected PHC clinics in response to the deinstitutionalisation of people with SMIs (NDOH, 2012). Implementation varies across the Sedibeng district in Gauteng province: some psychiatry services are "co-located" next to PHC clinics, while others are "physically integrated" into PHC (Abdulla et al., 2024). However, the extent to which collaboration between community psychiatry and PHC staff is achieved remains unknown. This study aims to explore healthcare providers' perceptions of the extent of this collaboration and its challenges by comparing different integrated care settings. We have published a mixed-methods study on HCUs' characteristics and experiences elsewhere (Abdulla et al., 2024).

## Background

While there are various descriptions of the CBCC model that vary by study settings and resources, the key components include multidisciplinary teams, regular communication and case management (see Table 1) (Whitfield et al., 2023). Multidisciplinary teams work together to provide comprehensive care for HCUs, guided by referral and stepped care protocols (Katon, 2009). Teamwork is achieved through various communication channels, including regular case review meetings, HCU progress letters and shared health records (Katon, 2012, 2009). Case management functions include coordination and continuity of care, HCU follow-up and progress tracking and delivery of behavioural health interventions (Dieterich et al., 2017). Case management functions are provided by individual case managers (generally supervised by psychiatrists) or available healthcare staff (Dieterich et al., 2017).

The levels of collaboration/integration framework categorises integration approaches to enable meaningful comparisons of the different approaches and their outcomes (Heath et al., 2013). Integration and collaboration occur in a continuum from no integration and minimal collaboration to integrated and full

**Table 1.** CBCC components

| Components | Description |
|---|---|
| Multidisciplinary team | Must include both:<br>• Mental health specialists (e.g., psychiatric nurses, psychologists and psychiatrists)<br>• PHC clinicians (e.g., family physicians, PHC nurses and medical assistants)<br><br>And at least one of the following categories of healthcare providers:<br>• Allied health care providers (occupational therapists, social workers, dieticians, etc.)<br>• Case manager(s)<br>• Non-professional healthcare providers (lay health counsellors and community health workers) |
| Communication | Systematic communication channels, including any of the following:<br>• Regular (weekly/monthly) case review meetings with at least two cadres of healthcare providers related to the HCUs' treatment and care<br>• Detailed referral/progress letters with follow-up communication<br>• Shared health records that contain longitudinal observations of all providers |
| Case management | Case manager or provider(s) who perform any of the following responsibilities:<br>• Follow-up on missed appointments<br>• Monitoring HCU care<br>• Coordinating input from various healthcare providers<br>• Delivering evidence-based behavioural health interventions |

*Source*: Katon (2012, 2009) and Dieterich et al. (2017).

collaboration (Heath et al., 2013). For this article, we adapted the framework focusing on two integration approaches: co-located and physically integrated. The framework categorises collaboration levels from basic collaboration to full collaboration (see Table 2) (Heath et al., 2013). While the framework focuses on collaboration between PHC and community psychiatry providers, we also use it to report on collaboration between community psychiatry staff.

## Methods

This study used a qualitative study design. Individual semi-structured qualitative interviews were conducted from October 2021 to May 2022 to explore healthcare providers' experiences of CBCC in integrated settings. This study is reported in accordance with the Standards for Reporting Qualitative Research guidelines (O'Brien et al., 2014).

### Setting

In the Sedibeng district, specialty community psychiatry services provide psychiatric care, treatment and rehabilitation for HCUs with SMIs, supporting PHC providers in delivering primary mental health care. The services are managed by the district mental health team, consisting of medical doctors, psychiatrists, psychiatric nurses, psychologists, occupational therapists and social workers. Due to insufficient providers in the district, community psychiatry staff (other than the psychiatric nurses) rotate between three and four clinics. The community psychiatry staff at each clinic work alongside PHC clinicians and allied providers. Figure 1 provides current referral practices in the district.

**Table 2.** Levels of collaboration/integration framework

| Co-located<br>In the same facility, but in separate buildings | | Physically integrated<br>In the same facility, within the same space | |
|---|---|---|---|
| Level 1: Minimal collaboration | Level 2: Basic collaboration | Level 3: Close collaboration | Level 4: Full collaboration |
| • Have separate systems<br>• Communicate regularly about shared patients, by phone or e-mail<br>• Collaborate, driven by the need for each other's services and a more reliable referral<br>• Meet occasionally to discuss cases due to close proximity<br>• Feel part of a larger yet non-formal team | • Share some systems, like scheduling or medical records<br>• Communicate in person as needed<br>• Collaborate, driven by the need for consultation and coordinated plans for difficult patients<br>• Have regular face-to-face interactions with some patients<br>• Have a basic understanding of roles and culture | • Some separate systems (e.g., filing systems) but actively seek system solutions together or develop workarounds<br>• Communicate frequently in person<br>• Collaborate, driven by the desire to be a member of the care team<br>• Have regular team meetings to discuss overall patient care and specific patient issues<br>• Have an in-depth understanding of roles and culture | • Functions as one integrated system<br>• Communicate consistently at the system, team and individual levels<br>• Collaborate, driven by a shared concept of team care<br>• Have formal and informal meetings to support an integrated model of care<br>• Have roles and culture that blur or blend |

*Source*: Adapted from Heath et al. (2013).

### Study sites

Two clinics, both formerly district general hospitals, were purposefully selected for this study. A detailed description of the two clinics is provided in Table 3.

### Data collection

Staff were purposively selected to participate in the study and included facility managers, administrators, psychiatrists, psychiatric registrars, psychiatric nurses, allied providers and PHC clinicians. Data were collected by the first author using the interview guide focusing on the working environment, referral process and collaboration with multidisciplinary providers. Five interviews with healthcare providers were conducted in a pilot site to standardise the interview guide and improve interviewer techniques. Pilot interviews were not analysed. Thereafter, a total of 29 interviews (16 in Clinic-1 and 13 in Clinic-2) were conducted, and data saturation was reached. All interviews were conducted in English, and audio recordings were transcribed verbatim by the first author and two PhD students. Interviews lasted between 25 and 75 min, excluding the consent process. Data were collected in-person or telephonically to accommodate participants' busy schedules or to ensure safety precautions during the coronavirus disease (COVID) pandemic.

### Data analysis

Interviews and field notes were manually analysed using thematic analysis (Braun and Clarke, 2006). Transcripts were re-read by the first author and at least one other author to increase familiarity with the data and to generate codes. Codes were grouped into themes by clinic and healthcare discipline. Emerging themes were collaboratively discussed during regular meetings until consensus was reached. Two central themes emerged from the data: the structural barriers to effective healthcare delivery and the extent of CBCC. To reduce interviewer bias and improve the credibility and reliability of the findings, the interviewer kept a reflective journal with field notes during data collection and analysis (Nowell et al., 2017).

### Findings

Out of the 29 participants, the majority were female (72%), had worked in the clinic for <2 years (69%) and provided care in multiple clinics (55%) (Table 4).

### Structural barriers to effective health care delivery

#### Infrastructure

In the physically integrated clinic, there was insufficient space for consultation and administration (such as paper-based filing systems). For example, the allied staff were located in a small container outside in the parking area: "*We are struggling with space. In these three rooms, there's speech and audiology, physio, and a podiatrist. Sometimes I wait outside so that this treatment area can be used for seeing the patient*" (Occupational therapist, Clinic-2). Staff also complained about the neglected infrastructure: "*I need a very dark room to do (eye) examinations… my windows are broken. I requested curtains when I got here … it's going to be a year next month*" (Optometrist, Clinic-2).

In the co-located clinic, the provision of community psychiatry services in the outbuilding of the PHC facility provided adequate space. However, the outbuilding was not maintained well: "*Mental health is neglected… the mental health [building] needs to be up to par with PHC building*" (Administrator, Clinic-1).

#### Safety and security

In both clinics, little was done to ensure staff safety: "*Security guards are on strike today. Even before that, security runs away. Very often, they're not even stationed at mental health*" (Psychiatrist, Clinic-1). Security guards were not trained to manage HCUs who had relapsed or were not stable: "*Security are afraid of the patient*" (Occupational therapist, Clinic-2). In the physically integrated clinic, there were additional challenges: "*Someone went through security and (then) stabbed a patient*" (Administrator, Clinic-2). Clinic management did not offer staff any support or debriefing sessions after these incidents.

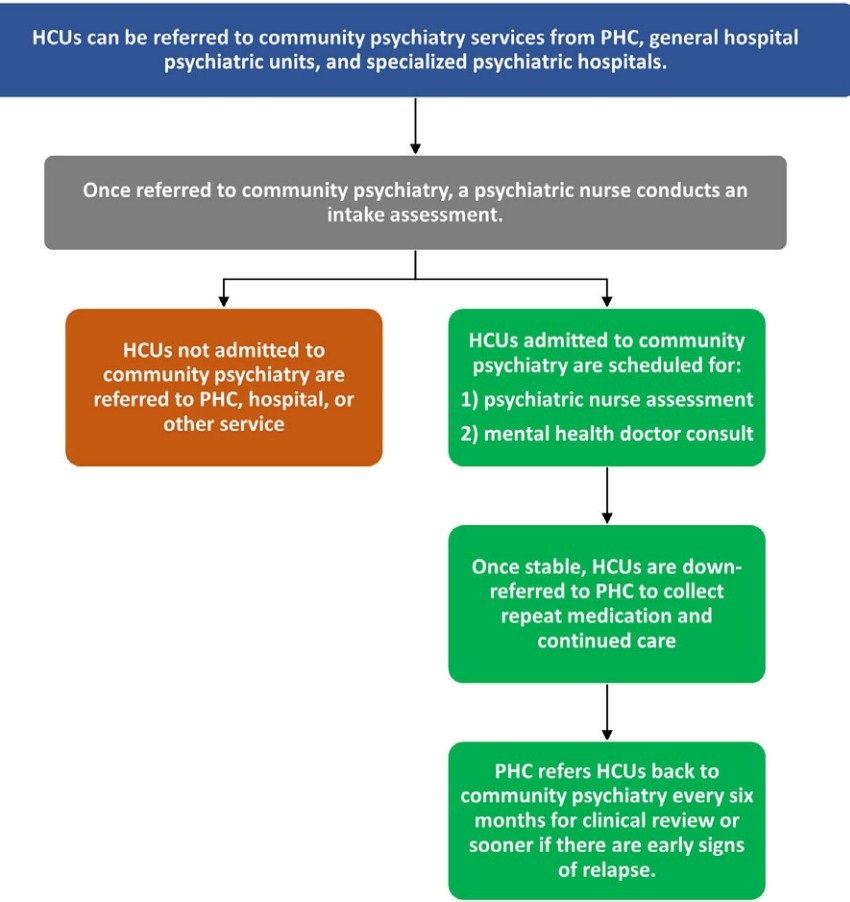

**Figure 1.** Referral process for HCUs requiring psychiatric care.

**Table 3.** Description of PHC and integrated community psychiatry services

|  | Co-located clinic (Clinic-1) | Physically integrated clinic (Clinic-2) |
|---|---|---|
| Catchment population | The clinic is in a suburban area and is accessed by a racially mixed population of mixed socio-economic status. | The clinic is in a township and accessed by Black South Africans of poor socio-economic status. |
| Physical space | PHC is in the main building. Community psychiatry is in an outbuilding behind PHC. | PHC and community psychiatry are in the same building, with limited space. |
| Filing systems (paper-based) | Separate clinical records and filing systems for community psychiatry and PHC. | Shared clinical records and filing system. |
| Service load in community psychiatry | ~580 HCUs | ~910 HCUs |
| Multidisciplinary team | Community psychiatry in both clinics has the same amount of allocated human resources that provide care to only HCUs with SMIs: Approximately five nurses are available daily on weekdays in each clinic. (Nurses work at the same clinic each day.) Each clinic has two to four doctors who provide psychiatric care once a week for adults, and once a week for children and adolescents. (The doctors rotate between clinics.) Psychologists, social workers and occupational therapists provide care bi-weekly. ||
| | PHC staff: Both clinics have a variety of staff including casualty doctors, general nurses and PHC allied staff (i.e., dieticians, optometrists, physiotherapists, general occupational therapists and social workers and speech and hearing therapists among others). PHC allied staff provide care biweekly, rotating between clinics. ||
| Physician roles | Community psychiatry doctors deliver outpatient psychiatric and general health care for uncomplicated medical comorbidities. ||
| | PHC doctors deliver more complex physical health care and after-hours mental health care. ||
| Nurse roles | Psychiatric nurses: Conduct mental health screening of new HCUs when they first access community psychiatry services, issue psychotropic medication and provide clinical follow-up for HCUs every month. ||

*(Continued)*

**Table 3.** (Continued)

| | Co-located clinic (Clinic-1) | Physically integrated clinic (Clinic-2) |
|---|---|---|
| | Psychiatric nurses focus primarily on mental health. | Psychiatric nurses provide PHC nursing and mental health duties. |
| | PHC nurses: Stable HCUs with SMIs are down-referred from community psychiatry to dedicated PHC nurses (i.e., mental health champions) in the PHC clinic who issue repeat medication (that are prescribed by psychiatric doctors every 6 months). | |
| Management structure | In theory, the facility manager oversees the PHC clinic and the community psychiatry service, but in practice, she only engages with PHC. | The facility manager oversees the PHC clinic and the community psychiatry service. |
| | All staff report to senior line managers in their specialty (e.g., psychologists report to a senior psychologist). | |

*Notes*: (1) Township refers to an area located on the outskirts of a city that was historically reserved for Black South Africans during the Apartheid era. (2) The racial categories used in this study are based on the South African government's classification system used in official statistical reports (Statistics South Africa, 2022).
*Source*: Adapted from Abdulla et al.( 2024).

**Table 4.** Characteristics of staff interviewed in both clinics[1]

| Staff | Sex | Years working at the clinic | Previous psychiatric work experience | Clinic rotations |
|---|---|---|---|---|
| Facility manager | 2F | 1–4 years | 1 Yes, 1 No | Full-time in one clinic |
| Administrator | 1F, 1M | 1–10 years | 2 No | Full-time in one clinic |
| Psychiatric nurses | 3F | 2–11 years | 1 No, 1 Yes, 1 unknown | Full-time in one clinic |
| Psychiatrist registrar | 2F | 2–3 months | 2 Not known | Three clinics |
| Psychiatric medical officer | 1F | 8 months | Not known | Three clinics |
| Psychiatrist | 1F | 4 years | Yes: extensive | Three clinics |
| Psychologists | 3F | 5 months to 1 year | Not mentioned | Three to four clinics |
| PHC physician | 1F | 2 years | No | Full-time in one clinic |
| Casualty physician | 1M | 6 years | Yes: limited | Full-time in one clinic |
| PHC community service doctors | 1M, 1F | 4–6 months | 2 No | Full-time in one clinic |
| Mental health champion | 1F | 9 years | No | Full-time in one clinic |
| Occupational therapists | 1F, 2M | 8–11 months | 3 No | Three clinics |
| Social workers' supervisor | 1M | 1 year | Yes: extensive | Four clinics |
| Social worker (community psychiatry service) | 2F | 4–9 months | Not known | Three clinics |
| PHC social worker | 1F | 6 months | Not known | Full-time in one clinic |
| Dietician | 1F | 5 months | No | Four clinics |
| Optometrist | 1M | 11 months | No | Three clinics |
| Physiotherapist | 1M | 3 months | No | Three clinics |

[1]Due to the specific study population, characteristics of staff interviewed were aggregated for both clinics to ensure anonymity of participants.

### Inadequate material resources

In both clinics, limited resources led to HCUs being referred to hospitals: "*You have to refer everybody out because there's little resources at PHC… equipment is always a problem*" (PHC Physician, Clinic-1). The physically integrated clinic was most under-resourced: "*If I want to do heat therapy, we don't have the resources at clinic-2, but it's available at clinic-1*" (Physiotherapist, Clinic-2). Insufficient resources were further exacerbated by high caseloads: "*We exhaust our budget very fast. We have already exceeded the number of people that we're supposed to service*" (Administrator, Clinic-2). Inadequate resources affected morale: "*Public health is exhausting in this country- you're not the type of doctor you want to be. You can have the knowledge, but because we don't have*

*equipment, we're so limited and then our patients are dying and it's on us*" (PHC community service doctor, Clinic-2).

The facility manager from the physically integrated clinic explained the problem: "*There's no support, no resources in the clinic. At district, it's worse. We don't have a chief that's directing our district. They keep giving us an acting chief for two months… who changes everything you were doing. Staff take out their frustration on me. Most of the time, there's no stock because providers were not paid, but we're expected to render a quality service…with what?*" (Facility manager, Clinic-2). Given the dire need for resources, the facility manager acquired stock from nearby clinics: "*I go to clinics to borrow [medical] supplies… They say: 'It's you again, what do you want now?'*" (Facility manager, Clinic-2).

### Human resources and workload

In both clinics, many posts were vacant: "*We need a chief occupational therapist and assistant director. We only have a deputy director*" (Occupational therapist, Clinic-2). In both clinics, the administrative staff contracts were not renewed: "*We have 8 admin staff who have one-year contracts. Last week their contracts lapsed, and the 5 permanent staff must now cover their work*" (Administrator, Clinic-2).

The COVID-19 pandemic increased both the caseload and the staff: "*With all the mental health awareness post-COVID, our patient burden is increasing, but now we also have allied health care professionals and more doctors. We're able to offer a better multi-disciplinary team approach to the management of mental health disorders*" (Psychiatrist, Clinic-1). Despite the increase in staff, the workload remained high, impacting collaboration: "*I don't know a lot of my colleagues. You're swamped with seeing patients back-to-back, and then you go home. I'm not even aware of some of the policies here*" (Psychologist, Clinic-2).

Staff also reduced their consultation times: "*…to see more patients…but you can miss subtle things like depression… I do my best to prioritize urgent issues, and then leave other issues for the next visit*" (Psychiatrist registrar, Clinic-2). This impacted the quality of care: "*If one nurse is sick or on leave, it's quite a strain to us because we're seeing about 60–70 patients per day, so we don't provide quality care*" (Psychiatric nurse, Clinic-2).

In both clinics, the overburdening of full-time staff (psychiatric nurses and PHC clinicians) led to conflict with district management: "*In December, we were ready to leave… we weren't getting support from the people higher than us. It's exhausting working over 24 hours. We argued a lot and finally got more staff, but now they want to take the doctors away because there's not enough rooms*" (PHC community service doctor, Clinic-2).

### Extent of CBCC

#### The role of leadership in facilitating collaboration

The physically integrated clinic operated under strong facility management, thus reducing conflict between staff across disciplines in the facility: "*When I arrived here, it was a mess. There were groups fighting. Being a psychiatric nurse helped me notice that this is not okay. I sat them down and said: 'We need to work this way. I'm going to support everyone, whether you like it or not'. They all ended up supporting me*" (Facility manager, Clinic-2). The facility manager encouraged shared responsibility among staff: "*I involve them. I told them I'm not only the manager, but you are also managers…*" (Facility manager, Clinic-2). The facility manager's approach enabled staff to address issues constructively: "*She's (facility manager) quite good. If there's any issues, we discuss appropriately*" (PHC community service doctor, Clinic-2).

In contrast, the co-located clinic was poorly managed: "*There's no proper hierarchy of whom to report problems to, or how the matter's dealt with. You don't feel like you belong*" (PHC dietician, Clinic-1). The facility manager had limited involvement in community psychiatry: "*They are not part of my organogram*" (Facility manager, Clinic-1). A member of the community psychiatry team described the problem: "*We are neglected. There's a burnt bridge between us and PHC facility management*" (Psychiatric nurse, Clinic-1).

The lack of management at the community psychiatry service in the co-located clinic resulted in a senior psychiatric nurse inappropriately stepping into a role of authority. Without the necessary leadership skills or approval from the district, this resulted in a hostile work environment, affecting team dynamics: "*If I ask about a patient file, or speak to one of the nurses, she (senior psychiatric nurse) shouts at them afterwards and tells them not to speak to me, even if it's for a patient*" (Occupational therapist, Clinic-1). Dealing with power dynamics was challenging for many staff: "*The most difficult thing right now is dealing with different personalities. Sometimes the same person can p\*\*\* you off*" (Psychiatric nurse, Clinic-1). Senior management did not address the power dynamics in community psychiatry: "*People complain about her [senior nurse] but no one can do anything. People will rather go work at a clinic that's farther away to avoid her*" (Administrator, Clinic-1). Unhealthy power dynamics became part of clinic culture, reducing job satisfaction: "*The seniors are telling us to just come here for your payslip… do the bare minimum. They really need to fix all the problems to retain staff and get staff to work as a team*" (Occupational therapist, Clinic-1).

### Communication between staff

#### Within community psychiatry

In both clinics, community psychiatry staff used various methods to communicate with each other. HCUs' clinical records were updated after each consultation to provide longitudinal information. Staff also communicated in-person or telephonically, especially for serious cases: "*If it's a high-risk patient, I get feedback either telephonically or when I see the allied professional, otherwise the feedback is in the file*" (Psychiatrist, Clinic-1). Joint consultations between community psychiatry staff were common: "*Like if a patient feels that his wife doesn't understand his condition and medication, and that's getting in the way of his compliance, then the doctor will explain why medication is important and I'll explain why psychology is important, and we'll explain why they work together… it's happened three times this week*" (Psychologist, Clinic-1).

Community psychiatry staff attended regular district-wide psychiatry meetings. The meetings fostered teamwork and collaboration: "*We work very closely with the occupational therapists, psychologists, and social workers. In psychiatry, you need them*" (Psychiatrist, Clinic-2). Staff presented and discussed challenging cases they encountered: "*We sit together and discuss what we're unsure of, and we get input from everyone*" (Psychologist, Clinic-1). This improved treatment plan: "*I had a discussion with the doctor about the treatment plan for a patient and it really helped. You become more aware of deeper issues which you might not have thought of. And then treatment becomes more effective*" (Occupational therapist, Clinic-1). The meetings also fostered learning among staff: "*I must say with the academic format, I've learned a lot about mental health care*" (Occupational therapist, Clinic-1).

Staff used the meetings to get support and build interpersonal relationships with each other: "*We also form friendships with other staff. Sometimes I'll find myself with the occupational therapist talking about a patient and it ends up becoming a debriefing session… So it's not always formal*" (Psychiatrist registrar, Clinic-2). These interactions reduced feelings of isolation: "*There's a lot of support and that keeps me going. I'm not alone*" (Psychiatrist registrar, Clinic-2).

However, the meetings were not without challenges. At times, the meetings led to conflict: "*When nurses try to criticise doctors, their supervisor speaks up for them. Then nurses turn on us (occupational therapists). Oh do they feast on us because we're on*"

*our own… our supervisor isn't always there so we're like a group of headless chickens*" (Occupational therapist, Clinic-1).

Some psychiatric nurses from the co-located clinic did not attend meetings. Other nurses attended, but did not always participate: "*When we're discussing the patient, they (nurses) don't always join us. The senior manager always says how she's had to deal with their resistance. They don't want to come to meet her, they don't want to have to do this or that*" (Occupational therapist, Clinic-1). Resistance among nurses was attributed to the hierarchical structure in the clinic: "*The general perceived hierarchy is that nurses are at the bottom, so they (nurses) try to put others down so that they can be with them at the bottom of the chain*" (Occupational therapist, Clinic-1).

### Between community psychiatry and PHC

In the physically integrated clinic, the shared filing system offered comprehensive progress updates of HCUs, thus facilitating knowledge exchange among the multidisciplinary team. The close proximity of staff enabled psychiatric nurses to approach PHC nurses: "*We communicate with them most of the time. If there's a problem I don't understand, they (PHC) sister listen to us*" (Psychiatric nurse, Clinic-2). There was also regular communication between some psychiatrists and PHC staff: "*Some PHC doctors will walk to mental health to get advice or say hi. Some doctors want to learn and they call me at least once a week*" (Psychiatrist, Clinic-1).

In the co-located clinic, the physical separation between mental health and PHC limited collaboration: "*We're so segregated space-wise, PHC doctors never come here to get our opinion. There's always been a lack of collaboration between us. I have yet to receive a phone call this year from the PHC doctors*" (Psychiatrist, Clinic-1). Separate filing systems also limited information exchange between the mental health and PHC services. Traditional hierarchy limited collaboration between the psychiatric nurse and PHC doctor: "*I don't call the doctor because we always end up in a fight. They say: 'You are just a nurse'. But if I have worked for a long time in a mental health, I'm more qualified and I can advocate for the patient. Some doctors don't want to listen to that*" (Psychiatric nurse, Clinic-1). Only one mental health provider approached PHC clinicians for advice: "*Sometimes I will pop by PHC and ask them for advice*" (Psychiatrist, Clinic-1).

In both clinics, mild mental health conditions and non-complicated SMIs should be treated by PHC clinicians; however: "*…that's not happening. When the patient needs to get their script reviewed, the patient is referred to mental health to see the psychiatrist, which is a waste of time because that patient is stable. Why must the [PHC] doctor not review this patient?*" (Facility manager, Clinic-2). Despite the facility manager's efforts in the physically integrated clinic, PHC providers resisted providing mental health care: "*I saw resistance in mostly [PHC] doctors, they say: 'this is a psych patient, I don't do this' and then refer immediately. Then I will say please treat, please do counselling, you still have those basic skills. They don't even do their assessments properly to exclude mental illness*" (Facility manager, Clinic-2). This resistance often led to PHC staff incorrectly referring HCUs to community psychiatry: "*The patient came in for a chronic peptic ulcer that was escalated by the slight stress, but the PHC doctor wrote mental health condition and referred to us*" (Psychiatric nurse, Clinic-1).

Before COVID-19, psychiatrists provided training to PHC staff in both clinics regarding the referral protocol and managing HCUs with mental health conditions. However, this was discontinued due to the pandemic: "*We do training, but this year I haven't done any training with PHC because they're so overwhelmed with COVID*" (Psychiatrist, Clinic-1).

In both clinics, the lack of communication between PHC and community psychiatry staff was attributed to PHC doctors' resistance to treating mental health conditions: "*PHC doctors will say: 'Why must I see this patient when their clinic (community psychiatry) is there?'*" (Facility manager, Clinic-2). Some providers did not feel equipped to provide quality care: "*The level of care that I would be able to provide is inferior to what they are getting*" (PHC community service doctor, Clinic-2). Others felt that collaborating with community psychiatry staff would increase the workload.

### Case management: a core CBCC component

Neither clinic had case managers. While all staff updated clinical records when consulting with HCUs, other case management functions (continuity of care, care coordination and monitoring and following up HCUs) varied based on staff capacity.

### Coordination and continuity of care

In both clinics, some staff coordinated care with each other (i.e., PHC community service doctor, mental health champion, psychiatric nurses, social workers and occupational therapists): "*My patient told me that she's not taking medication, so I took her to psychiatry to arrange to see the doctor*" (Mental health champion, Clinic-1).

Senior psychiatrists provided continuity of care for high-risk HCUs: "*They are either booked to see me or, if I'm not there, the (mental health) doctor that sees them will phone me. It is more difficult to maintain a similar continuity of care for patients who are low-moderate risk*" (Psychiatrist, Clinic-1). In the physically integrated clinic, psychiatric nurses also helped to ensure continuity of care for high-risk HCUs: "*Nurses will ensure that the same doctor sees that high-risk patient. Some nurses do it very well… they know their patients*" (Psychiatrist, Clinic-1). In the co-located clinic, nurses did not provide continuity of care: "*There's just one pile of files. So, when each doctor's done with the patient, then you go and collect the next file at the top*" (Psychiatrist, Clinic-1). Some psychiatrists ensured continuity between each other: "*Another doctor will see my handwriting in another file, so she'll just bring that file to me and then I'll see the patient, so we do that amongst ourselves… unless we need a second opinion*" (Psychiatrist registrar, Clinic-2).

### Monitoring and following up of HCUs

In both clinics, psychiatric nurses (and the mental health champion in the co-located clinic) monitored the HCUs' progress when they attended the clinics for their monthly medication. However, those who missed their appointments were not followed up with.

In contrast, psychologists, occupational therapists and social workers contacted HCUs who missed their appointments. However, some HCUs slipped through the cracks: "*Today, the line was hectic. If 20 people defaulted today, it would be very difficult to call all those people. Some (HCUs) don't have a phone*" (Psychologist, Clinic-1). Given that PHC and mental health doctors did not conduct any follow-ups, they recommended having a dedicated person to do this task. Staff also recommended electronic systems to manage HCUs: "*With HIV (patients), we have an (electronic patient management) system that will tell you that the patient should have been at the clinic but didn't come. We need that for mental health patients*" (Facility manager, Clinic-1).

## Discussion

This study explores the extent of collaboration among staff in two integrated clinics in South Africa. A full multidisciplinary team was available in each clinic. While there was no case manager in either clinic, some staff conducted case management functions on an ad hoc basis. Both clinics had inadequate infrastructure, limited human resources and high workloads. However, the physically integrated clinic, despite having less space, fewer medical resources, a higher patient load and additional safety concerns, had better collaboration. A summary of the levels of collaboration and integration in the clinics is presented in Table 5.

In both clinics, community psychiatry staff attended regular district-wide psychiatry meetings, fostering teamwork, learning and shared decision-making, despite occasional resistance from some nurses and conflict between staff. In both clinics, there was **Level 4: full collaboration** among community psychiatry staff. Staff operated in the same space and shared filing systems, which facilitated knowledge exchange. Close proximity enabled regular communication between staff members. In the physically integrated clinic, the facility manager, with an interest and experience in mental health, played a proactive role in encouraging a positive organisational culture. However, in the co-located clinic, the facility manager's limited involvement in community psychiatry led to a senior nurse assuming an authoritarian role. This created a hostile work environment. Despite these challenges, joint consultations and communication still occurred. Interpersonal relationships built during regular district meetings may have helped mitigate the negative effects of these power dynamics.

Collaboration between PHC and community psychiatry staff varied in both clinics. In the **physically integrated clinic**, there were some elements of **Level 4: full collaboration,** where staff shared space and systems and communicated with each other. Despite some conflict over human resources and occasional resistance from PHC doctors to treat HCUs with mental health conditions, the facility manager's efforts in improving team dynamics mitigated the negative effects of hierarchy. In the **co-located clinic**, there was **Level 1: minimal collaboration** between PHC and community psychiatry staff. Separate spaces and systems hindered collaboration, resulting in limited collaboration between PHC and community psychiatry staff. Poor management reinforced traditional hierarchies and power imbalances, leading to unclear reporting structures, limited support for staff and minimal communication.

The integration of mental health in PHC settings is important for improving quality care for people with multi-morbidities (Zezai et al., 2024). Studies in LMICs have found that integrating mental health services into PHCs resulted in improved HCU outcomes, reduced stigma and improved access to care (Hanlon et al., 2020, 2022; Smith et al., 2020). However, there are limited studies on the experiences of CBCC for SMIs in integrated care settings in LMICs. For example, a systematic review on healthcare providers' experiences of CBCC for SMIs identified only two studies conducted in LMICs that met the inclusion criteria (i.e., had at least two of the three CBCC components) (Abdulla et al., 2025). Both studies, conducted in India (Pereira et al., 2011) and China (Li et al., 2020), found that the consistent and respectful efforts at engagement by psychiatrists led to gradual endorsement and buy-in from

**Table 5.** Levels of collaboration/integration in clinics

| Clinics elements | | Between the community psychiatry staff | Between the PHC and the community psychiatry staff |
|---|---|---|---|
| Co-located clinic | Level of collaboration | Level 4: Full collaboration <br> • Same space <br> • Shared filing systems and medical records <br> > > Shared HCU information <br> > > In-person or telephonic communication for serious cases <br> > > Occasional joint meetings | Level 1: Minimal collaboration <br> • Separate space <br> • Separate systems <br> > > Minimal communication <br> > > Community psychiatry staff do not feel part of the PHC team |
| | Strengths | Staff communicate with each other despite a hostile work environment | |
| | Weaknesses | • Limited facility management involvement <br> • Senior nurse taking power <br> • Intimidation tactics limit access to the HCU's information <br> • Leadership failure <br> > > Conflict/power dynamics <br> > > Traditional hierarchies remain | • Power imbalances and traditional hierarchies remain the same <br> • Staff are not supported <br> • No clear reporting structures <br> • Lower cadres (psychiatric nurses) are unable to discuss care with higher cadres (PHC doctors) <br> • Community psychiatry staff isolated from larger PHC team |
| Physically integrated clinic | Level of collaboration | Level 4: Full collaboration <br> • Same space <br> • Shared filing systems and medical records <br> > > Shared HCU information <br> > > In-person or telephonic communication for serious cases <br> > > Occasional joint meetings | Level 4: Full collaboration <br> • Same space <br> • Shared filling systems and treatment records <br> > > Shared HCU information <br> > > Some staff communicate in-person due to close proximity of services. |
| | Strengths | • No evidence of conflict between staff <br> • Good management reduces the negative effects of hierarchy <br> > > Mutually supportive relationships | The facility manager's efforts result in: <br> > > Improved organisational culture <br> > > Reduced negative effects of hierarchy (e.g., staff conflict) |
| | Weaknesses | | • Conflict with management regarding human resources and workload <br> • PHC doctors resist treating mental health despite the manager's efforts |

PHC doctors, in turn contributing to positive staff interactions (Pereira et al., 2011; Li et al., 2020). These studies also identified regular multidisciplinary team meetings as key to fostering communication between PHC and psychiatry providers. Similarly, a review on common mental disorders found that regular case meetings facilitated collaboration between providers and emphasized the importance of physical proximity between providers to enable collaboration (Overbeck et al., 2016). In our study, the implementation of systematic communication channels and team meetings between PHC and community psychiatry staff could improve teamwork.

There is a notable difference in the integration approach employed in various studies. In studies using the MhGAP guidelines to integrate care in Ethiopia (Hanlon et al., 2020, 2022), Nepal (Jordans et al., 2019), Uganda (Nakku et al., 2019), Rwanda (Smith et al., 2020) and Kenya (Mutiso et al., 2019), a task-sharing approach is used, where non-specialist providers deliver integrated mental health services. In other studies conducted in India (Pereira et al., 2011), China (Li et al., 2020), Kenya (Jenkins et al., 2013) and Uganda (Wakida et al., 2019), PHC doctors are primarily responsible for treating mental health conditions, whereas psychiatrists play supportive and supervisory roles. Similarly, in our study, PHC providers should theoretically manage HCUs with mild and moderate mental health disorders, as well as stable HCUs with SMIs, whereas community psychiatry staff should manage more complex HCUs. However, in both clinics, PHC doctors did not understand and/or resisted their role in delivering mental health services based on the presence of a district mental health team. Resistance in providing mental health care or even physical care to HCUs with mental health conditions contributed to the limited communication between PHC and community psychiatry staff. In the co-located clinic, the integration of community psychiatry services also created confusion in the management structure, resulting in limited involvement of the facility manager in overseeing community psychiatry. This may also have been reinforced by the physical separation of community psychiatry and PHC services in separate buildings. This emphasizes the importance for staff and management to have a clear understanding of the purpose of integration and their roles to improve collaboration in integrated care settings.

Strong leadership is essential for improving collaboration among staff and is considered a good indicator of a robust PHC system (Zezai et al., 2024). In the India study (Pereira et al., 2011), regular feedback from the intervention team facilitated staff engagement. In the China study (Li et al., 2020), endorsement of the CBCC intervention by village leadership and facility management led to buy-in and team engagement. In our study, we found that resistance from nurses (and PHC doctors) hindered collaboration. Studies in South African hospitals attributed similar resistance to staff being overworked, lacking resources, performing tasks beyond their scope of practice and mistrust of authority rooted in the legacy of apartheid, leading to poor communication and conflict (Fana and Goudge, 2021, 2024). Poor management led to resistance, low morale and disengagement, while strong leadership and clear communication fostered trust, loyalty and cooperation (Fana and Goudge, 2021). Good managers overcame resistance by acknowledging staff contributions and involving them in decision-making (Fana and Goudge, 2021). However, senior management roles often remain vacant, or insufficiently skilled people are appointed to senior posts, which affects team dynamics and staff retention.

Case managers play a significant role in facilitating communication between PHC providers and psychiatrists (Pereira et al., 2011). While neither clinic had a designated case manager, our study found that some staff performed case management functions on an ad hoc basis, including coordinating care, ensuring continuity and following up with HCUs who missed appointments. However, monitoring of community psychiatry HCUs during medication collection was conducted routinely. The absence of case managers to coordinate care between PHC and community psychiatry staff is a barrier towards achieving full collaboration. Considering the shortage of skilled healthcare specialists in LMICs, non-professional cadres can be upskilled to provide case management for SMIs (Pereira et al., 2011; Li et al., 2020).

## Limitations

This study has several limitations. This study used a qualitative design and was conducted in two clinics, limiting the statistical generalisability of the findings and potentially reducing the identification of the full range of factors that influence CBCC in integrated settings. However, qualitative research aims to achieve theoretical generalisability, providing in-depth findings that can be transferred to similar settings (Carminati, 2018). Selecting two facilities provided a detailed understanding of CBCC, thus contributing to theoretical generalisability and informing future research. Telephonic interviews reduced the length and depth of some interviews. In addition, high workloads prevented some staff from participating. In the co-located clinic, the interviewer was introduced to some staff members by a senior manager, which may have led to some nurses refusing to participate and potential bias in the responses of some staff members. However, the deep introspection from the staff suggests this may not have significantly impacted the results.

## Conclusion

Integration does not guarantee collaboration between staff. While all the elements of full collaboration were not achieved in either setting, the physically integrated setting (due to shared files, physical proximity and good management with mental health interest and experience) provided a better opportunity for communication among staff. However, these advantages were still hindered by poor infrastructure and inadequate resources. Improving collaboration between mental health and PHC staff requires facility managers who can support mental health integration, more human resources to reduce the workload and deliver case management functions and systematic communication channels (such as case review meetings) between PHC and community psychiatry staff.

**Open peer review.** To view the open peer review materials for this article, please visit http://doi.org/10.1017/gmh.2025.10020.

**Data availability statement.** The data supporting the conclusions of this article will be made available by the authors upon reasonable request.

**Acknowledgements.** The authors are thankful to the Centre for Health Policy and South African Research Chairs Initiative (SARChI) of the National Research Foundation, School of Public Health, Faculty of Health Sciences, University of the Witwatersrand, Johannesburg, South Africa.

**Author contribution.** All authors contributed to the study conception, design and analysis. As a manager in the Sedibeng District Health Services, Lesley Robertson did not have access to the raw data and was recused from the data analysis to preserve participant anonymity. Saira Abdulla is the primary manuscript writer. Lesley Robertson provided expert guidance. All authors contributed to and approved the submitted manuscript.

**Financial support.** This review is part of a broader PhD study funded by the South African Research Chairs Initiative (SARChI) of the National Research Foundation (NRF).

**Competing interests.** The authors declare no competing interests.

**Ethics statements.** Ethical approval for this study was obtained from the University of the Witwatersrand, Human Research Ethics Committee and Sedibeng District Health Services.

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
