## [Reviewer Report]

This is a well written paper on a specific but nevertheless interesting aspect of delivery of mental healthcare for severe mental illness in South Africa. It could be improved by more clearly situating the research within the existing evidence base, some suggestions below.

Introduction

- Suggest citing more recent literature on disease burden and WHO strategy as current references are 20 years + old. E.g. WHO Mental Health Action Plan 2013-30 (which also headlines integrated/ collaborative care). The WHO’s MhGAP is a widely implemented approach to integration of mental health care in primary care which could also be mentioned (including relevance, or not, to this setting and/or research).

Methods

- Suggest adding a reference for the Standards for Reporting Qualitative Research guidelines

Results

- Characteristics are given for individual participants. Despite the fact the clinics are not named, given this is a very specific study population, the authors might want to consider presenting in summary format to ensure individual participants are not identifiable.

- The overarching themes are ‘clinic environment’ and ‘community-based collaborative care’, and sub-themes are in a similar format e.g. ‘leadership’. These are not self-explanatory. Suggest renaming themes as short self-explanatory sentences to align with Braun and Clarke’s meaning of the term (“an idea or concept that captures and summarises the core point of a coherent and meaningful pattern in the data”)

- In some subthemes (e.g. ‘Inadequate material resources’) it is not always clear whether or how the findings related to the delivery of mental healthcare. This could be made clearer.

- Table 5 is a nice overview of the results. However in the ‘Between community psychiatry staff’ column the co-located clinic is labelled as Level 2: Basic Collaboration, and the Integrated clinic as Level 4: Full collaboration. Yet the details/notes in the cells are identical in the two clinic types. The authors could make clearer within the table what distinguishes the two clinic types in terms of collaboration (i.e. why is one considered Level 4 vs Level 2)

- In terms of the ‘Between PHC and community psychiatry staff’ column, the co-located clinic is categorised as Level 1: Minimal collaboration in the table, and Level 2: Basic collaboration in the text. These should be aligned.

Discussion

- The discussion feels quite narrow in scope. To expand the scope the authors could consider (i) citing literature on the effectiveness of mental health care delivered in primary care in LMIC e.g. Hanlon et al 2022 https://pubmed.ncbi.nlm.nih.gov/34921796/ (ii) reflecting on the importance/ relevance of integrated care for people with multiple long term conditions, citing relevant literature e.g. Zezai et al 2024 https://pubmed.ncbi.nlm.nih.gov/39608990/ (iii) reflecting on implementation challenges for integrated/collaborative care interventions for common mental disorders in primary care in South Africa (e.g. Petersen et al 2023 https://pubmed.ncbi.nlm.nih.gov/37956110/) and if any of that learning is relevant to SMI

- An important limitation is this study did not seek the views of service users or caregivers- this could be reflected upon in the discussion

---

## [Reviewer Report]

This manuscript reports on a qualitative investigation of two approaches (physically integrated and co-located) to integrated mental health care in two primary health care facilities in the Sedibeng district of Gauteng in South Africa. The study is limited by conducting the study in only two PHC facilities and while there are efforts to link the findings to global mental health literature in the discussion, conclusions are limited by the design.

---

## [Reviewer Report]

This is an interesting and timely study in the context of deinstitutionalized care and integrated, collaborative community-based care.

1. The title is misleading and suggests that this is a large South African study. Suggest specifying the scope and location of the study.

2. Line 59, p3 Suggest adding a more recent reference

3. Line 76, p3 Are the authors suggesting that the studies are unreliable? It would be helpful to clarify why these are considered “lower-quality” and why these are included if unreliable

4. Line 83, p3 Province is mentioned earlier. Are the authors referring to province or district. If district, this should be made clearer i.e. in one district or across # districts. Terminology should be consistent.

5. Line 132, p6 -PHC is repeated

6. Line 133, p6 Figure 1 is unclear, blurry and text not visible. Suggest higher resolution picture

7. Line 22, p9 Were either of the patients HCUs with SMI?

8. Line 285, p1 "can piss you off” may be considered vulgar in terms of academic publishing- consider rephrasing with * or the word expletive. At editorial discretion

9. Line 332, p12 The table indicates the presence of two nurses. Using the description “some” and “others” implies a larger number. Suggest rephrasing for accuracy -i.e. One nurse or sometimes ....

10. Line 434-435 This statement suggests a larger study than the 2 facilities reported

11. Line 443 -Table 5 “Intimidation tactics to limit access to HCU information”. This assertion is not supported with a quotation either and should be made clearer in the quotations.

12. Line 492, p17 Suggest “based on” or another phrase/word that indicates the reason for the resistance was the close proximity of the mental health teams rather than “considering”

13. Line 503, p17 “Strong leadership are essential” is grammatically incorrect. There are other minor grammatical errors not highlighted. Recommend authors review and edit where necessary

14. Limitations section

This study has a limited sample of 1 clinic in each category with a total of 2 clinics. There may be several limitations as a result including not taking into account various contextual and other factors that may be unique to these clinics, and that may have led to the reported conclusion. A larger sample may have confirmed the results or may have revealed multiple other factors for consideration. Recommend this section and conclusion include these and any other limitations and resulting conclusions

---

## [Editor Report]

Dear Ms Abdulla

We have received reports from three peer reviewers, and, based on the points raised in these reports, the decision on your submission is a request for a Major Revision. This is largely due to a discordance between the methods employed and the inferences drawn. While this would normally lead to a Rejection, there value in interrogating the consequneces of PHC integration models, though this would require adding additional data to better support the conclusions of the paper. It is recommended that the research team consider adding quantitative measures and more clinics in order to provide more robust evidence for the differences across models. In light of the practical consequences of such a decision, this might not be completely feasibile though, in which case it is recommended that the team consider submitting the paper to an alternative journal. Regardless of your decision, we would like to thank you for submitting to GMH, and look forward to future submissions.